# Measuring implicit sequence learning and dual task ability in mild to moderate Parkinson´s disease: A feasibility study

**Malin Freidle**[1]*, **Hanna Johansson**[1,2], **Alexander V. Lebedev**[3], **Urban Ekman**[4,5], **Martin Lövdén**[6,7], **Erika Franzén**[1,2,8]

**1** Division of Physiotherapy, Department of Neurobiology, Care Sciences and Society, Karolinska Institutet, Stockholm, Sweden, **2** Allied Health Professionals Function, Function Area Occupational Therapy & Physiotherapy, Karolinska University Hospital, Stockholm, Sweden, **3** Department of Clinical Neuroscience, Karolinska Institutet, Stockholm, Sweden, **4** Division of Clinical Geriatrics, Department of Neurobiology, Care Sciences and Society, Karolinska Institutet, Stockholm, Sweden, **5** Allied Health Professionals Function, Medical Unit Medical Psychology, Karolinska University Hospital, Stockholm, Sweden, **6** Aging Research Center, Department of Neurobiology, Care Sciences and Society, Karolinska Institutet, Stockholm, Sweden, **7** Department of Psychology, Gothenburg University, Gothenburg, Sweden, **8** R&D Unit, Stockholms Sjukhem, Stockholm, Sweden

* malin.freidle@ki.se

**Data Availability Statement:** Anonymized data can be found at the study's OSF page: osf.io/x9baq/ DOI 10.17605/OSF.IO/X9BAQ For legal and ethical concerns, we are confined to sharing the data with

## Abstract

We investigated the feasibility aspects of two choice reaction time tasks designed to assess implicit sequence learning and dual task ability in individuals with mild to moderate Parkinson's disease in comparison to healthy individuals. Twelve individuals with mild to moderate Parkinson's disease and 12 healthy individuals, all ≥ 60 years of age, were included. A serial reaction time task was used as a measure of implicit sequence learning and a similar task but with the addition of a simple counting task, was used as a measure of dual task ability. We have present thorough descriptive statistics of the data but we have refrained from any inferential statistics due to the small sample size. All participants understood the task instructions and the difficulty level of both tasks was deemed acceptable. There were indications of task fatigue that demand careful choices for how best to analyse the data from such tasks in future trials. Ceiling effects were present in several accuracy outcomes, but not in the reaction time outcomes. Overall, we found both tasks to be feasible to use in samples of individuals with mild to moderate Parkinson's disease and healthy older individuals.

## 1. Introduction

Parkinson's disease (PD) is an age-related neurodegenerative disorder. It is mainly characterized by progressive loss of dopamine cells in the substantia nigra and lowered levels of dopamine available for neurotransmission. The disease encompasses a wide range of motor and cognitive symptoms such as rigidity (muscle stiffness), bradykinesia (small, slow movements) and deficits in balance and gait, executive functions, and memory [1, 2]. Research also indicates deficits in motor learning and dual-tasking i.e., the simultaneous performance of two tasks, in individuals with PD [3–5]. The deficits in motor learning that individuals with PD

group belonging (Parkinson - healthy) as the sole demographic information to ensure complete anonymity. All data used for descriptive statistics and graphs of behavioral performance is however be openly shared.

**Funding:** E.F. Swedish Research Council, https://www.vr.se, 2016-01965 E.F. Parkinson research foundation, https://parkinsonfoundation.se, grant number not applicable/used E.F. Centre for Innovative Medicine for financial support, https://cimed.ki.se, grant number not applicable/used The funders had no role in study design, data collection and analysis, decision to publish, or preparation of the manuscript.

**Competing interests:** The authors have declared that no competing interests exist.

experience, might make a large contribution to the disabilities in gait, balance and dual tasking, plausibly due to a diminished ability to adapt to changing and demanding conditions [4].

Many motor tasks, such as reaching for and grabbing a cup, consist of complicated sequences of movements (lifting the arm, moving the hand towards the cup, opening and fixating the fingers around the cup's handle etc.) [6]. The learning and adaption of motor sequences can be both explicit i.e., with directed attention and awareness, and implicit i.e., without directed attention and awareness. Evidence suggests that implicit task learning taxes working memory to a lesser extent than explicit task learning and that motor skills learned implicitly are possibly less affected by anxiety or secondary tasks than motor skills learned explicitly [7–9]. With these benefits of implicit learning, it is important to note that implicit learning is reported to be more negatively affected than explicit motor learning in individuals with PD [4]. It is possible that a deficit in implicit learning is an important cause of the motor difficulties that are characteristic of PD.

The serial reaction time task (SRTT) developed by Nissen and Bullemer [10], is commonly used to investigate implicit sequence learning. It is a type of a computer-based choice reaction time task where some trials follow a repeating hidden sequence.

A meta-analysis of SRTT studies reported that individuals with PD show about half a standard deviation lower implicit sequence learning than healthy older individuals. The authors emphasised however, that small and heterogenous samples as well a diversity in study design might have obscured important patterns in the results [11].

Choice reaction time tasks have also been used to measure dual task ability through the addition of a secondary task. Older healthy individuals tend to have an impaired dual task ability in comparison to younger individuals [12] and it has frequently [3, 13], but not always [14], been reported that this impairment is amplified in individuals with PD.

A large randomized controlled trial (RCT) by our research group showed that a highly challenging balance training positively affected gait and balance as well as the dual task performance of a cognitive task (performed while walking) in individuals with mild to moderate PD [15, 16]. We are now planning a similar trial with the addition of brain activity assessments as measured by functional magnetic resonance imaging (fMRI), see our study protocol [17]. During fMRI, we plan to use a version of the SRTT as a measure of implicit sequence learning ability and to use a similar choice reaction time task with the addition of a simple counting task (from here on referred to as the dual task), as a measure of dual task ability. Because the tasks will be performed during the acquisition of brain activity data, we took extra precautions to minimize movement during the task performance as movement substantially lowers the quality of the brain activity data.

The aim with the present study was to evaluate feasibility aspects of the SRTT and the dual task in a sample of elderly individuals with mild to moderate PD and elderly healthy individuals. Feasibility assessments of task design, task delivery, and statistical analyses will help optimise future use of the tasks.

Because we did not have the statistical power for reliable inferential statistics, we did not perform any hypotheses testing. However, to allow the reader access to the data used to assess the feasibility of the two tasks, as well as to enable inclusion of our data in future meta-analyses, we thoroughly report the descriptive statistics and have shared the data set, scripts of the statistics and the task design files.

## 2. Methods

### 2.1 Procedure

The two tasks were performed in a quiet setting. The tasks were preceded by a training session where the participants first practiced the SRTT with only random trials. This training ended

when the participant achieved 80% accuracy or after a maximum of three rounds. The training also included four rounds of the dual task (no early termination of the training). To avoid task fatigue, the training was followed by a break (1–2 minutes), the SRTT was divided in two parts with a break in-between (~10 minutes) and there was also a break in-between the SRTT and the dual task (~20 minutes). The total time to perform the protocol ranged from 60 to 90 minutes depending on the participant's characteristics. All participants received two movie vouchers as compensation after participation. After providing oral and written study information, written consent was obtained from all participants. The study was approved by the regional Research Ethics Board of Stockholm (2016/1264–31/4 with amendments).

## 2.2. Participants

In this cross-sectional feasibility study, 15 individuals with mild to moderate PD and 13 healthy individuals were recruited through advertisement in local newspapers and by contacting individuals who had previously showed interest in or who had participated in studies by our research group. Inclusion criteria for individuals with PD were a diagnosis of idiopathic PD, age $\geq$ 60 years, a score $\geq$ 21 on the Montreal Cognitive Assessment (MoCA) [18], Stage 2 or 3 on the Hoehn & Yahr scale [19] and no disease or symptom, over and above PD, that could significantly affect balance or gait (same inclusion criteria used as for the upcoming RCT). Inclusion criteria for healthy individuals were age $\geq$ 60 years, a MoCA score $\geq$ 23 and no disease or symptom that could significantly affect balance or gait. A cut–off of 23 on MoCA showed excellent sensitivity and specificity for mild cognitive impairment in a sample of healthy older adults [20]. The lower MoCA cut-off for the participants with PD was chosen to recruit a representative sample of the PD population in which cognitive deficits are common. Participants with PD were tested in the ON state of dopaminergic medication. There was no fixed time for testing in relation to the last medication intake because the participants differed in type of dopaminergic medication used and thereby in the medications' duration of action. Two individuals with PD were not included due to MoCA < 21, and one individual with PD was excluded due to an administrative error. One healthy individual was not included due to MoCA < 23. All in all, 12 individuals with PD and 12 healthy individuals were included. The individuals with PD and the healthy participants did not differ substantially in age (p = 0.59) or education (p = 0.95). See Table 1 for more detailed descriptive data of the included participants.

## 2.3 Task designs

The tasks were presented in Psychopy (version 1.85.4) on a laptop with a 15-inch screen. In both tasks, the participants selected and pressed buttons depending on the stimuli presented on the screen. They used the index and middle finger of both hands and two two-button response pads, one for each hand, see Fig 1.

**Table 1. Sample characteristics.**

| | Age years | Sex | Education years | MoCA | Hoehn & Yahr Stage 2/3 | Years with diagnosis | MDS–UPDRS III: Motor exam. |
|---|---|---|---|---|---|---|---|
| | *mean (sd)* | *m/f* | *mean (sd)* | *mean (sd)* | *mean (sd)* | *mean (sd)* | *mean (sd)* |
| PD (*n* = 12) | 70.3 (6.2) | 9/3 | 15.3 (2.6) | 25.8 (2.0) | 8/4 (2.33) | 7.2 (6.7) | 34.6 (10.0) |
| Healthy (*n* = 12) | 71.4 (2.8) | 5/7 | 15.2 (3.6) | 26.8 (2.5) | n/a | n/a | n/a |

Note: PD = Parkinson's disease, MoCA = *Montreal* Cognitive Assessment, MDS-UPDRS III: Motor exam. = the Movement Disorders Society-Unified Parkinson's Disease Rating Scale, Part III: Motor examination, n/a = not applicable.

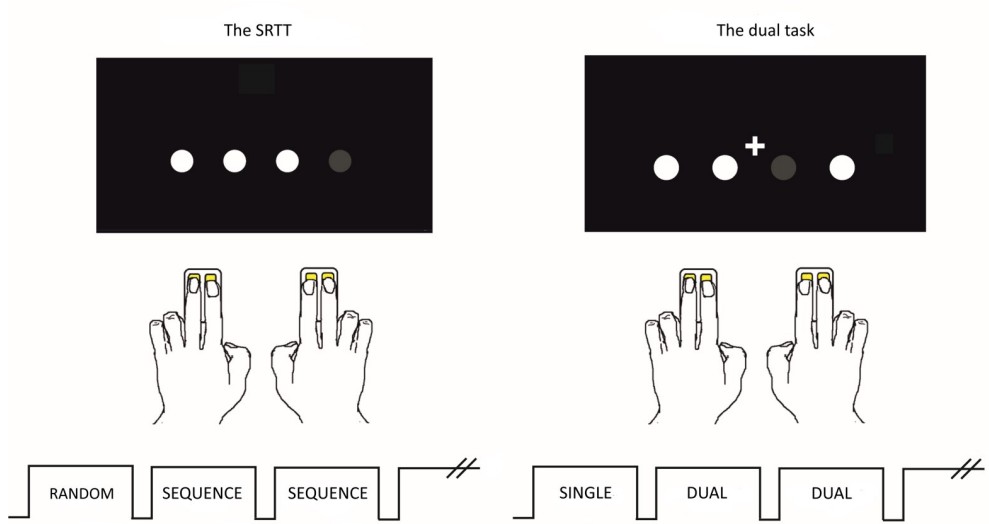

**Fig 1. Set-up of the SRTT and the dual task.** Left: The SRTT. Interleaved random and sequence blocks. The differences in reaction time and percent accuracy between the last two random blocks and the last two implicit blocks, evaluated before and after a break, were used as measures of implicit sequence learning. Right: The dual task. Interleaved single and dual task blocks. The differences in reaction time and percent accuracy between the single blocks and the dual blocks as well the performance of correctly counting the plus signs, were used as measures of dual task ability.

For the SRTT, four white circles on a horizontal line were shown on a black screen and each circle's position corresponded to one of the four buttons of the two response pads. Every 1.2 seconds, one of the circles turned grey and the participants were to press the corresponding button as quickly as possible. The SRTT consisted of 19 blocks of 40 trials each, and all blocks were followed by a six second break. Unbeknownst to the participants, in 12 of the 19 blocks the trials followed a 10–item higher (second) order sequence (1 3 1 2 4 3 2 4 1 3), from here on referred to as sequence blocks. Participants were exposed to the sequence 48 times (24 times before the break). The remaining seven blocks consisted of trials presented in a random order with the constraint that a trial could not be the same as the directly preceding trial. These blocks will from here on be referred to as random blocks. The break in the SRTT was placed after the first ten blocks.

In the dual task, the participants were again instructed to look at the four circles and as quickly as possible press the button corresponding to the circle that turned grey. The task consisted of eight blocks and to prevent fatigue, the blocks were shorter (32 trials) than in the SRTT. All blocks consisted of trials presented in a random order but in five of the blocks, a plus sign briefly showed up just above the circles five to ten times per block (Fig 1, right side). In these dual task blocks, the participants were to count the plus signs, as well as press the buttons corresponding to the circles turning grey. To make the participant aware of the approaching dual task and to avoid surprise effects and accompanying movement, each dual block started with all four circles briefly turning green. After each block, the participants had six seconds to respond to how many plus signs they had seen by choosing between four alternatives (ranging from 0 to 7 integers from the correct alternative). This non-verbal multiple-choice question was used instead of a free recall to minimize the risk of movement when the task is to be performed in the MRI scanner in our upcoming trial [17]. The participants were instructed to focus on both tasks to an equal extent.

After performing both the SRTT and the dual task, the participants were asked to fill out a questionnaire on whether they thought the trials in the SRTT followed a pattern. This was to

assess whether any participants had achieved explicit knowledge of the sequence, which would lead to exclusion of their data in the analyses of the SRTT. If the participants had any notion of a pattern, they were asked to reproduce this pattern on a sheet of paper where the first trial of the sequence was given. A score (maximum of six points) was derived from questions about the participants' experience of (a) the non-random nature of the trials ("I believe the grey circles occurred in random locations, I did not notice a pattern" (0 points), "I think the locations of the grey circles could have followed a pattern, but I am unsure" (1 point), "I am pretty sure the locations of the grey circles were not random, but I am not sure what the pattern was" (2 points), "I am pretty sure the locations of the grey circles were not random, and I think I know what pattern they followed" (3points), "I am sure the locations of the grey circles were not random, and I am sure I know what pattern they followed"(4 points)) and (b) the description of the regularity across blocks ("A pattern was repeated over and over again" (1 point), "A pattern was occasionally repeated" (2 points), "Some locations occurred more often than others" (0 points).

## 2.4 Descriptive statistics

Data from the SRTT for one healthy participant could not be used due to technical problems and data of one participant with PD were not included due to severe diplopia during the task performance. For four of the participants with PD, one of the response pads did not work. This was not noticeable during the experiment and all descriptive statistics and graphs include their responses from the working response pad. However, these participants were not included in the descriptive statistics and graphs of the counting task where responses from only one response pad would be unreliable due to the small number of total responses. For both the SRTT and the dual task, only correct responses, and responses with a reaction time (RT) > 100ms were included. 100ms has been reported to be the minimum time for physiological processes such as the perception of stimuli [21]. Non-responses were coded as incorrect responses. The software R 3.6.2 [20] was used for all statistics.

**2.4.1 Descriptive statistics of the SRTT.** As a measure of implicit sequence learning, we calculated the differences in performance between the last two random blocks and the last two sequence blocks before the break (the mean of blocks 7 and 10 vs. the mean of blocks 8 and 9) and after the break (the mean of blocks 16 and 19 vs. the mean blocks 17 and 18). The median RT and the mean accuracy were first calculated per block and per individual, then averaged over the pairs of blocks described above, and then used for the group level statistics. The differences between the random blocks and sequence blocks in relation to the sequence blocks were calculated in percent as $\left( \frac{(RT_{random} - RT_{sequence})}{RT_{sequence}} \right) * 100$ and $\left( \frac{(Accuracy_{sequence} - Accuracy_{random})}{Accuracy_{sequence}} \right) * 100$. For the participants with only one working response pad, the percentage of accuracy were calculated in relation to the number of possible correct responses for the working response pad (20 each for the four sequence blocks, 20 and 23 for the two random blocks before the break and 19 and 18 for the two random blocks after the break). In addition to reporting the outcomes of the SRTT as a measure of implicit sequence learning, we also took use of the data by separately calculating percent accuracy, median RT, and semi-interquartile range (SIQR) for just the random trials for the two groups.

**2.4.2 Descriptive statistics of the dual task.** For the dual task, the median and the SIQR of the individuals' RTs were calculated for the single and dual blocks separately. The mean of the medians was then calculated separately for type of block (single and dual) and for the two groups. We calculated the difference in RTs and correct response for the single blocks compared to the dual blocks in percent as $\left( \frac{(RT_{dual} - RT_{single})}{RT_{single}} \right) * 100$ and $\left( \frac{(Accuracy_{single} - Accuracy_{dual})}{Accuracy_{single}} \right) * 100$.

These differences will from here on be referred to as dual task costs. The added task of counting the plus signs was evaluated in two ways 1) by the number of times that the correct alternative was chosen and 2) through a scoring system. Within the scoring system, a participant received three points if the correct alternative was chosen and one point if the alternative closest to the correct alternative was chosen. The other alternatives rendered zero points.

### 2.5. Feasibility aspects

The following feasibility aspects were assessed: 1) if the participants reported that they understood the instructions for the tasks and if there were any other individual-related obstacles to task execution, 2) if there were indications of ceiling or floor effects (measured as very high or low accuracy levels), 3) if there were indications of task fatigue (measured as the difference in RT and accuracy between the first and the last random block before and after the break in the SRTT and the first and last block of the dual task), 4) if the difficulty level of the tasks was reasonable (measured as whether most trials were responded to, the means and range of accuracy levels, and if any participant declined or could not complete the tasks) and 5) if the participants showed explicit awareness of the sequence in the SRTT (measured by the questionnaire and pattern reproduction test described under "2.3 Task designs")?

In addition, we wanted to investigate whether our task designs would show indications of implicit sequence learning for at least the healthy group and a dual task cost for both groups as these patterns have been found in earlier research [3, 12, 22]. Because motor slowness is a core symptom of PD, we were also curious to investigate whether the individuals with PD showed a pattern of worse performance on the random trials than the healthy participants. Consequently, further feasibility aspects assessed were: 6) if there were indications of implicit sequence learning for the group of healthy individuals (measured as a pattern of lower RTs and higher accuracy for the implicit trials than for the random trials), 7) if there were indications of implicit sequence learning present already before the break, 8) if there were there indications of a dual task cost in both groups (measured as a pattern of higher RTs and lower accuracy in the dual blocks compared to the single blocks), and 9) if the PD group showed a pattern of higher RTs and lower accuracy on the random trials of the SRTT than the healthy group. The purpose of investigating these feasibility aspects was to guide us in potential modifications of the task designs, and we want to emphasise that the quantitative results are not robust due to our small sample size.

### 2.6 Reproducibility statement

An anonymised version of the data, the scripts for the statistics and all files needed to run the tasks can be downloaded at osf.io/x9baq/ (doi:10.17605/OSF.IO/X9BAQ).

## 3. Results

### 3.1. Descriptive statistics of the SRTT

For the PD group, the mean RT of the random blocks was 6% higher than the mean RT of the implicit blocks before the break (the mean of blocks 7 and 10 vs. the mean of blocks 8 and 9), and 5% higher after the break (the mean of blocks 16 and 19 vs. the mean of blocks 17 and 18). For the healthy group, the mean RT of the random blocks was 7% higher than the mean RT of the implicit blocks before the break and 5% higher after the break.

For the PD group, the mean accuracy of the random blocks was 4% lower than the accuracy of the implicit blocks before the break (the mean of blocks 7 and 10 vs. the mean of blocks 8 and 9) and 3% lower after the break (the mean of blocks 16 and 19 vs. the mean of blocks 17

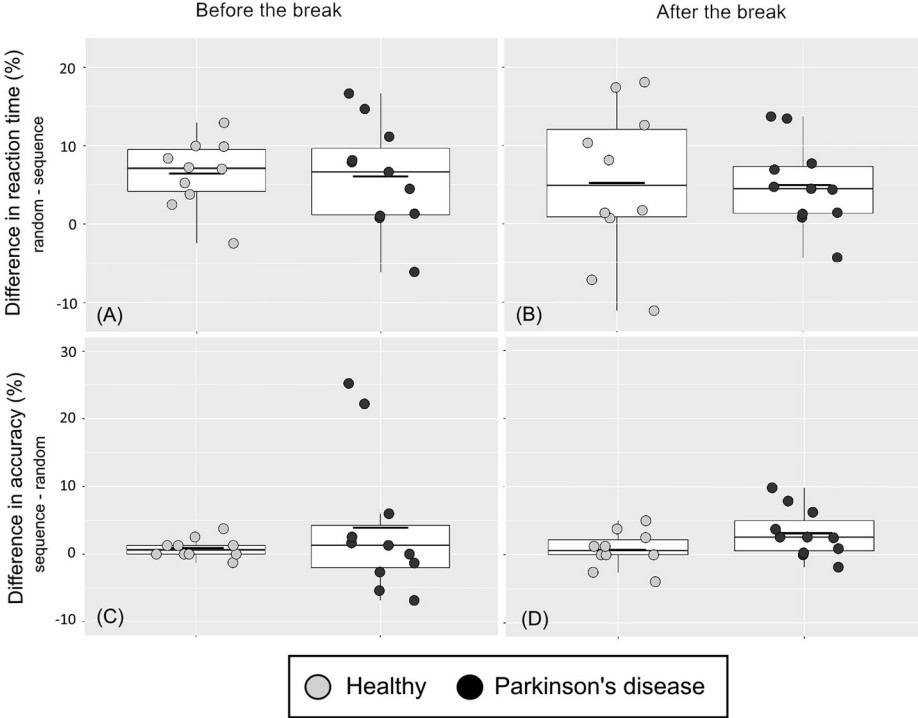

**Fig 2. Implicit sequence learning outcomes of the SRTT on the group and individual level.** (A) and (B): A value above zero indicates implicit sequence learning as measured by lower RTs for the last two sequence blocks compared to the last two random blocksbefore (left) and after (right) the break. (C) and (D): A value above zero indicates implicit sequence learning as measured by higher accuracy for the last two sequence blocks than for the for last two random blocks before (left) and after (right) the break. Each boxplot shows the median (the thin line), the mean (the short bold line), the 25th and 75th percentiles (the hinges), and the range from the smallest to the largest value if not exceeding 1.5 times the interquartile range (the whiskers).

and 18). For the healthy group, the mean accuracy of the random blocks was 1% lower than for the implicit blocks before the break, as well as after the break. See Fig 2 for the differences in RT and accuracy values of all individuals and see Fig 3 for the RT and accuracy outcomes over all blocks.

For the random trials, the mean RTs (of the individuals' medians) before the break were 580ms (*sd* = 76ms) for the PD group and 526ms (*sd* = 71ms) for the healthy group. The mean RTs (of the individuals' medians) for the random trials after the break were 565ms (*sd* = 111ms) for the PD group and 503ms (*sd* = 65ms) for the healthy group. In terms of intra-individual variance, the mean SIQR was 112ms for the PD group and 81ms for the healthy group before the break and 108ms for the PD group and 69ms for the healthy group after the break.

The mean accuracy (of the individuals' means) for the random trials before the break was 91% (*sd* = 9 percent units) for the PD group and 98% (*sd* = 2 percent units) for the healthy group. The mean accuracy (of the individuals' mean) for the random trials after the break was 91% (*sd* = 9 percent units) for the PD group and 98% (*sd* = 2 percent units) for the healthy group. See Fig 3 for the mean values for RT and accuracy over all blocks of the SRTT.

### 3.2 Descriptive statistics of the dual task

The mean RT (of the individuals' median) of the dual blocks was 24% higher than that of the single blocks in the PD group and 19% higher in the healthy group. The mean accuracy of the

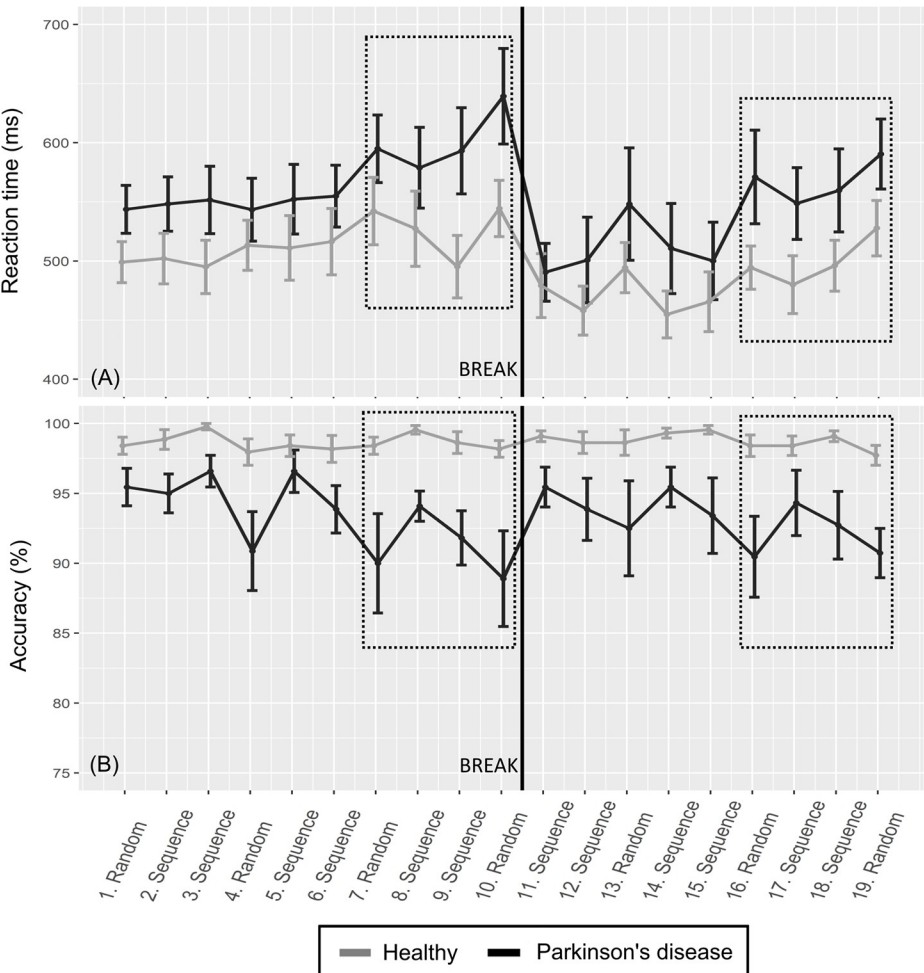

**Fig 3. Reaction time and accuracy of the SRTT.** (A) Means and standard errors (+1,−−1) of the RTs over all blocks for the two groups. (B) Means and standard errors (+1,−−1) of accuracy over the all blocks for the two groups. The squares indicate the blocks used for the descriptive statistics of implicit sequence learning where the performance on the random blocks was compared with the performance on the sequence blocks.

dual blocks was 8% lower than that of the single blocks for the PD group and 2% lower in the healthy group. See Fig 4 for dual task cost estimates and performance on the counting tasks for all individuals. See Fig 5 for the group mean RT and accuracy over all blocks. For intra-individual variance, the mean SIQR for the dual blocks was 126.6 ms for the PD group and 93.3 ms for the healthy group and the mean SIQR for the single blocks (of the dual task) was 99.3 for the PD group and 64.6 for the healthy group.

### 3.3. Feasibility evaluation

**3.3.1. Primary feasibility aspects.** All included participants understood the instructions but a few spontaneously expressed a wish for a longer training session. However, all participants were able to perform the tasks except for one participant with PD who experienced severe diplopia during both tasks. The healthy group showed indications of a ceiling effect in the accuracy outcome in both the random and implicit blocks of the SRTT (all mean values of the blocks were > 98%). The healthy group also showed indications of a ceiling effect in the accuracy outcomes in the dual task (all mean values of the dual blocks were ≥ 94%, and all

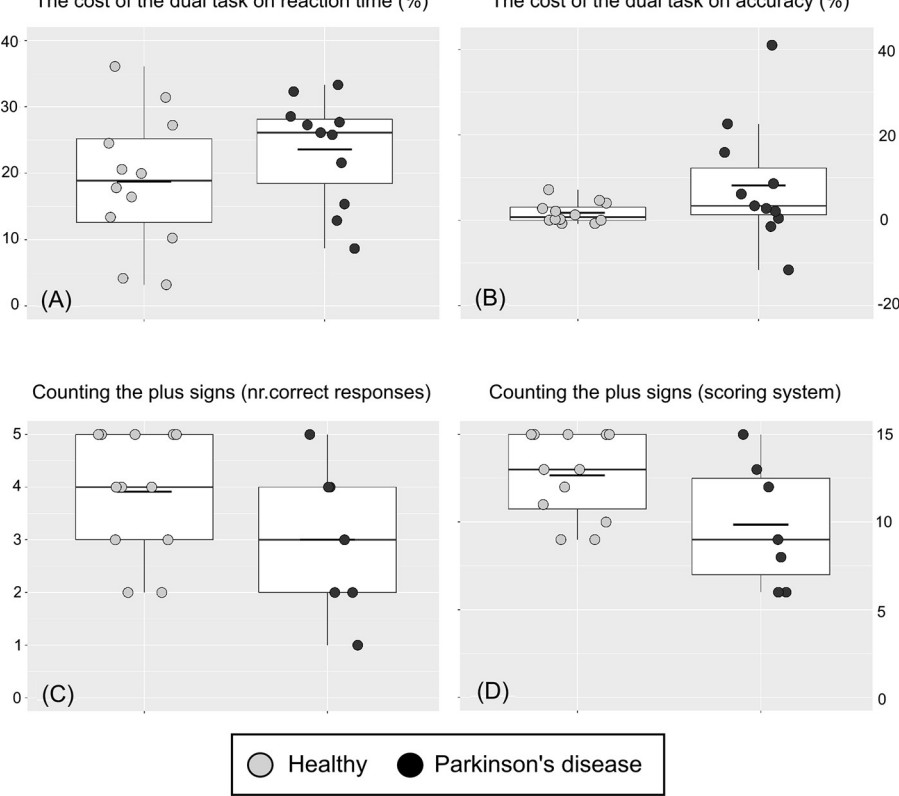

**Fig 4. Outcomes of the dual task with performance on the group and individual level.** (A) and (B): A score higher than zero indicates a cost of the dual task for that individual/group and outcome. (C): Performance on the counting task with one point given for each correct answer with a maximum value of 5. (D): Performance on the counting task by a scoring system (3 points for the correct answer, 1 point for the answer closest to the correct answer and 0 points for other answers) with a maximum value of 15. Each boxplot shows the median (the thin line), the mean (the short bold line), the 25th and 75th percentiles (the hinges) and the range from the smallest to the largest value if not exceeding 1.5 times the interquartile range (the whiskers).

mean values of the single blocks were $\geq$ 98%). There were no signs of floor effects in the SRTT or in the dual task.

Concerning task fatigue in the SRTT, there was an increase in RT of 96ms for the PD group and 45ms for the healthy group from the first random block to the last random block before the break. After the break there was an increase of 42ms for the PD group and 33ms for the healthy group from the first random block to the last random block. There was a decrease of 7 percentage units in the point estimates of accuracy from the first random block to the last random block for the PD group before the break, but no difference was seen for the healthy group. After the break there was a decrease of 2 percentage units for the PD group and 1 percentage unit for the healthy group. In the dual task, there was an increase in the point estimates of RT from the first block to the last block of 23ms for the PD group and 8ms for the healthy group. There was a decrease of 1 percentage unit in accuracy from the first block to the last block of the dual task for both groups.

For the difficulty level of the SRTT, the median of non-responses was 0% for the healthy participants before the break, but two participants did not respond for some random trials (range 2.5–2.5%). The median of non-responses for participants with PD was 1.9% before the break, and ten participants did not respond for some random trials (range 0.6–12.2%). After the break, the median of non-responses for the healthy participants was again 0%, but two of

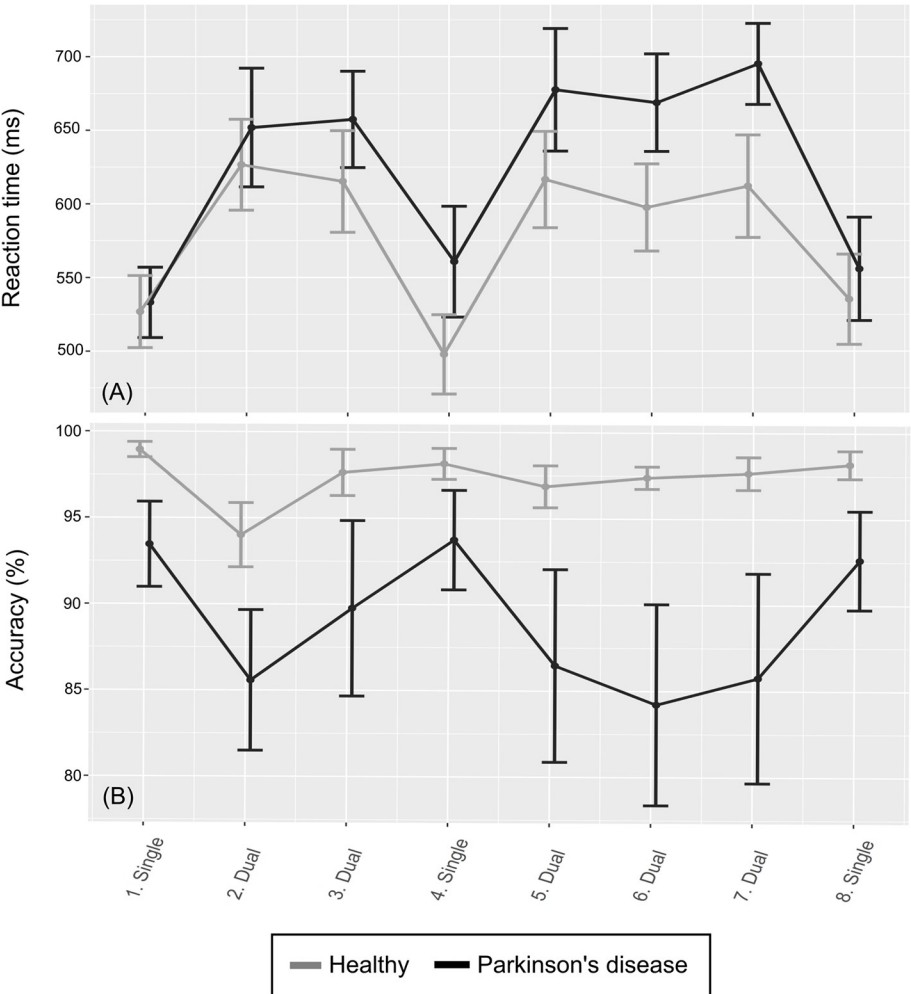

**Fig 5. Reaction time and accuracy of the dual task.** (A): Means and standard errors (+1, –1) of RTs over all blocks for the two groups. (B): Means and standard errors (+1, –1) of accuracy over all blocks for the two groups.

them did not respond to some of the random trials (range 1.7–2.5%). The median of non-responses for the participants with PD was also 0%, but five did not respond to some of the random trials (range 1.7–12.5%). All participants completed the SRTT. For the dual task, the median of non-responses for the dual trials was 0.6% for the healthy participants and nine of them did not respond to some dual trials (range 0.6–6.9%). The median of non-responses to dual trials for the participants with PD was 2.1% and ten did not respond to some dual trials (range 1.9–20.6%). See "3.2 Dual task ability" for the means and ranges of the accuracy levels of the button pressing part of the dual task and see Fig 5 for summary values and ranges for the counting task outcomes. All participants completed the dual task.

Table 2 shows the awareness of the sequence in the SRTT. The participant who correctly reproduced two triplets of trials was excluded from all descriptive statistics of implicit sequence learning.

**3.3.2. Expected patterns in the data.** There were indications of implicit sequence learning because the within-group mean RTs were larger for the random blocks than the implicit blocks, for both groups and before and after the break in the SRTT. The RT outcomes of the dual task indicated a dual task cost for both groups. However, the median values of the dual

Table 2. Awareness of the sequence in the SRTT.

| | No pattern | Could be a pattern, unsure | Might be a pattern, do not know pattern | Think there is pattern, probably know pattern | There is a pattern, know the pattern | Correct reproduction of triplets of pattern (number of triplets) | Awareness score, mean (sd) |
|---|---|---|---|---|---|---|---|
| PD (*n* = 11) | 4 | 2 | 4 | 0 | 0 | 0 | 1.8 (1.7) |
| Healthy (*n* = 11) | 4 | 2 | 5 | 1 | 0 | 1 (2) | 2.3 (2.1) |

task cost accuracy outcome (Fig 5, upper right) were close to zero for both groups. There was also a pattern of slower and less accurate response on the random trials of the SRTT for the PD group compared to the healthy group.

## 4. Discussion

In this study, we investigated the feasibility of two tasks designed to primarily assess implicit motor sequence learning and dual task ability in elderly individuals with mild to moderate PD and in elderly healthy individuals. All in all, the feasibility of the two tasks was supported but considerations in relation to task fatigue and ceiling effects for some accuracy outcomes are important for future use of the tasks.

All participants reported that they understood the task instructions, but because a few participants expressed a wish for a prolonged training session, we recommend increasing both the single and the dual task training by one or two training blocks. Because one participant could not perform the task accurately due to diplopia, and because visual difficulties are common in PD [23], we recommend future trials to pay attention to problems in vision that could prevent a fair evaluation of task-related abilities.

Ceiling effects, as observed in several accuracy outcomes in this feasibility study, result in inaccurate estimations due to reduced variation [24]. Without ceiling effects, higher accuracy of the implicit blocks than the random blocks in the SRTT might be possible and could be interpreted as implicit learning of the sequence. However, due to the ceiling effect in accuracy for the random blocks shown by the healthy group, a higher accuracy for the implicit blocks is not possible, and thus accuracy cannot be used as a measure of implicit sequence learning for the healthy group or when comparing the two groups. Due to the ceiling effect in accuracy in the single blocks of the dual task for the healthy participants (all single blocks $\geq$ 98% accuracy), accuracy also cannot be used as a measure of dual task cost for the healthy group or when comparing the two groups. Therefore, if the tasks are used with healthy older individuals, assessments should be made on RT outcomes rather than accuracy outcomes.

The indication of task fatigue that was especially visible in the SRTT raises the question of how to best analyse the data. A common design of the SRTT is to have only one random block, inserted as the last block of the task and then to compare the RT of this block with the RT of one or several preceding implicit blocks [11]. However, in such a design the RT of the random block might be higher in comparison to the implicit blocks partly due to task fatigue i.e., it might not solely be due to implicit learning of the sequence. We included the two last blocks of each type, before and after the break separately, to counteract fatigue as a confounder of implicit sequence learning ability. However, in a higher-powered trial where hypothesis testing is possible, another option would be to model all implicit and random trials over the time of the task. The fatigue effect is less problematic for the dual task as well as for the random trials of the SRTT because summary measures of all trials over the blocks of interest were used. We do, however, strongly recommend a break with the possibility to relax both the eyes and the fingers, when the two tasks are to be performed consecutively.

The difficulty level was satisfactory in that all participants could perform the tasks but as seen by the observed ceiling effects, an optimal difficulty level for all outcomes of a task is difficult to achieve. Some of the participants with PD showed quite large numbers of non-responses, especially on the dual trials, but the majority of the participants had low numbers of non-responses (all median values for both tasks < 2.5%). A slower presentation pace of the trials would probably decrease the non-response level, but it would also either prolong the task, with increased risk of task fatigue, or necessitate fewer trials and thereby less data. A slower pace could also mask important differences between the two groups. Because the SRTT consisted of 760 trials (400 before the break) and the dual task consisted of 256 trials, a 2.5% non-response level still gives RT and accuracy data for a large number of trials. We also believe that the counting task is on an acceptable difficulty level because there was a quite large variation in the number of accurate responses. It is possible that a more complex dual task would have resulted in larger and more robust effects on performance. However, the reaction of some of the participants when the task was presented to them, suggests that a more complex task could have increased anxiety and decreased willingness to perform the task. With this background, we deem the difficulty level of the two tasks to be acceptable for individuals with mild to moderate PD without severe cognitive problems and for healthy older individuals. However, future trials might make use of sensitivity analyses where individuals with high levels of non-response are excluded. Similarly, participants with a low accuracy level can also be excluded from analyses of implicit learning because low overall accuracy makes it difficult to estimate the learning of the sequence.

The patterns of implicit sequence learning and dual task costs presented here cannot be considered robust due to our small sample size, but they lend support to the design of the tasks and we see no reason to modify either the SRTT or the dual task based on these outcomes. However, because the pattern of implicit sequence learning was present already before the break in the SRTT, we recommend this shorter version because it diminishes the risk for task fatigue. The mean awareness scores were similar to those of a study using the same questionnaire [22]. We deem it acceptable that one participant showed awareness of the hidden sequence, and it is also plausible that fewer repetitions of the sequence–as is the case in the shorter SRTT version–will further prevent participants' awareness of the sequence.

A limitation of the present study was the technical problem with a response pad that decreased the amount of data for the affected participants. However, we believe that we used a defensible way to include the remaining data. We also acknowledge the possibility of erroneous button presses due to difficulties in motor control and severe tremor in some of the individuals with PD. It is, however, our experience that the response pads we used required a significant force that prevented registration of many involuntarily, light presses, and we also counteracted errors caused by involuntarily movements by excluding trials where the RT was ≤100 ms.

In addition, one participant had to be excluded due to an administration error. This participant was unable to understand the task instructions for the SRTT and by our mistake we did not continue with other assessments such as MoCA or UPDRS. We cannot rule out that this participant would have passed our inclusion criteria, including ≥ 21 points on the MoCA, but we deem this unlikely based on the difficulties shown. For more feasibility aspects, it would have been optimal if the participants in the present study could have performed the tasks inside a mock scanner in order to simulate the MRI experience in the forthcoming study, but this was unfortunately not possible. However, equivalent response pads as used in the present study are commonly used for both training and for in-scanner use. Because PD is a neurodegenerative disease where symptoms develop over time, we strongly believe that our restriction to include individuals within a limited range of disease severity (Hoehn & Yahr 2–3) is a great

strength. Large disease stage heterogeneity can confound the analyses of deficits and abilities that may be disease-stage dependent, and we look forward to the research field maturing in this respect.

The present study evaluated the feasibility of two tasks aimed at investigating implicit sequence learning and dual task ability in individuals with mild to moderate PD in comparison to healthy individuals, all $\geq$ 60 years of age. We deem the overall feasibility to be supported for both tasks but with important considerations primarily regarding task fatigue and ceiling effects. We have reported a wide range of descriptive statistics of the outcomes of the investigated tasks but due to our lack of statistical power we have refrained from any claims of reliable conclusions on abilities and deficits.

## Acknowledgments

We would like to thank all of the participants. We also thank Petra Koski for administrative assistance and Matthew Hogg at the Semantix Språkcentrum AB for professional writing assistance.

## Author Contributions

**Conceptualization:** Malin Freidle, Hanna Johansson, Alexander V. Lebedev, Urban Ekman, Martin Lövdén, Erika Franzén.

**Data curation:** Malin Freidle.

**Formal analysis:** Malin Freidle.

**Funding acquisition:** Erika Franzén.

**Investigation:** Malin Freidle, Hanna Johansson.

**Methodology:** Malin Freidle, Hanna Johansson, Alexander V. Lebedev, Urban Ekman, Martin Lövdén, Erika Franzén.

**Project administration:** Erika Franzén.

**Supervision:** Alexander V. Lebedev, Martin Lövdén, Erika Franzén.

**Validation:** Malin Freidle.

**Visualization:** Malin Freidle.

**Writing – original draft:** Malin Freidle.

**Writing – review & editing:** Hanna Johansson, Alexander V. Lebedev, Urban Ekman, Martin Lövdén, Erika Franzén.

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
