## [Decision Letter · Decision Letter 0]

2 Feb 2021

PONE-D-20-28422

Measuring implicit sequence learning and dual task ability in mild to moderate Parkinson's Disease: A feasibility study

PLOS ONE

Dear Dr. Freidle,

Thank you for submitting your manuscript to PLOS ONE. After careful consideration, we feel that it has merit but does not fully meet PLOS ONE’s publication criteria as it currently stands. Therefore, we invite you to submit a revised version of the manuscript that addresses the points raised during the review process.

We look forward to receiving your revised manuscript.

Kind regards,

Imre Cikajlo, Ph.D.

Academic Editor

PLOS ONE

Journal Requirements:

2.Thank you for including your ethics statement: 

"Written consent given by the regional Research Ethics Board, Stockholm. 2016/1264-31/4 with amendments".   

4.We note that you have indicated that data from this study are available upon request. PLOS only allows data to be available upon request if there are legal or ethical restrictions on sharing data publicly. For more information on unacceptable data access restrictions, please see http://journals.plos.org/plosone/s/data-availability#loc-unacceptable-data-access-restrictions.

Additional Editor Comments:

Please review the manuscript according to the reviewers' suggestions.

Reviewers' comments:

Reviewer's Responses to Questions

**Comments to the Author**

1. Is the manuscript technically sound, and do the data support the conclusions?

Reviewer #1: Yes

Reviewer #2: Yes

2. Has the statistical analysis been performed appropriately and rigorously? 

Reviewer #1: N/A

Reviewer #2: N/A

3. Have the authors made all data underlying the findings in their manuscript fully available?

Reviewer #1: Yes

Reviewer #2: Yes

4. Is the manuscript presented in an intelligible fashion and written in standard English?

Reviewer #1: Yes

Reviewer #2: No

5. Review Comments to the Author

Reviewer #1: PLOS one review

This study was a nice feasibility study that looked at implicit versus explicit motor learning in people with Parkinson’s Disease. To determine the influence of cognitive demands on motor learning, the authors included a dual task to a common key-press experiment. The study was intended to confirm the feasibility of using the task inside an fMRI machine. The authors have promoted transparency by providing their data and analysis, and used descriptive statistics to draw inferences because their study was admittedly underpowered.

I think this is a great feasibility study, and I think more studies like this should be published. I just have minor suggestions for clarity before publication. Also please note there are some grammatical errors throughout the manuscript that should be addressed by an editor and this reviewer will ignore.

Line 37: RT hasn’t been previously defined

Line 47: Are you referring to deficits in motor learning and dual-tasking specific to PD? Please clarify. There is a modern meta-analysis that you could cite in your intro:

Raffegeau, T.E., Krehbiel, L.M., Kang, N., Thijs, F.J., Altmann, L.J.P., Cauraugh, J.H., Hass, C.J., 2019. A meta-analysis: Parkinson’s disease and dual-task walking. Park. Relat. Disord. 62, 28–35. https://doi.org/10.1016/j.parkreldis.2018.12.012

Line 50: The following sentence is vague, please clarify that you are referring to people with PD.

Line 52: Please clarify what you mean when you say walking consists of a ‘series of complicated movements’. Adding a sentence or two would help clarify the rationale of the experiment. I believe the argument could be that movement becomes less automatic with PD and thus could cause interference in motor learning mediated by cognitive processes.

Line 79-83: Please clarify briefly that fMRI imposes limitations on cognitive test administration that this study is designed to overcome. A novice reader may be lost without a brief review of the fact that the key task has to be administered in an isolated way without the participant moving, etc.

Line 101-102: Add medication state information to the Participants section 2.2. Please add if you controlled the study such that your testing time was during optimal medication states, i.e. 1-2 hours after taking their medication?

I cant quite tell by the figure but I believe by the description of the task is that the keys are button-press keys. This may seem quite minor, but this reviewer is curious is the keys were rigid enough such that the commonly reported hand tremors wouldn’t influence performance for a PD patient? How much force is required to register a result? Later, there is good detail about how they examined key presses and excluded what appeared to be random responses, my guess is that if the key were too easy to press, that a tremor could be registered as a response.. A major strength of this design would be therefor the capability to prevent random key presses that are the result of bradykinesia or dyskinesia. For instance, a ‘touch pad’ key that doesn’t incorporate any haptic feedback may not be a good idea because it would pick up many more false positives. This may be worth mentioning in the Methods and Discussion.

Later, given the interest in task fatigue, I found myself curious as to how long the protocol took and how long the break was, but I don’t see that information? Maybe I missed it, but highlighting that would help interpret your results.

Line 212: Starting this sentence with a letter and numbers is confusing, adding the word ‘R software’ to this sentence would help.

The section 2.5 is a nice detailing of the goals of the study. The Results are pretty hard to follow as is, but this numbering system helps. Can the authors please incorporate this numbering system to section 3.3? It might be nice to have a section 2.4.1 for SRRT descriptives and 2.4.2 for dual task descriptives, that would help the reader follow the corresponding Results in section 3.1 and 3.2.

If you have enough room it would be nice to have a table or figure that presents the survey results, rather than reading through a long section.

A suggestion for all figures, which do a nice job of showing individual data points. I suggest you use the ‘jitter’ function in whatever program you are using so that your data points re randomly distributed rather than displayed in ‘rows’

Line 394-396: I agree with this point and I think also that the rate of learning would be an interesting outcome for your paradigm. Perhaps moving away from magnitude based hypothesis testing to testing the slope of the change would be best, especially in complicated clinical samples.

Line 407-409: While I agree your counting task likely leads to sufficient cognitive demand for a feasibility study, you could more carefully load ‘cognitive processes’ rather than simply dividing attention by adding some sort of inhibition/switching component to the task, which would greater challenge cognitive control/executive functions. While the counting task does load simple cognitive processes, your results could be more robust to PD-related deficits if you incorporated broader networks of cognitive processing. For instance, changing the color of the crosses periodically and instructing participants to only count Red crosses. Or changing the shape of the symbol and asking them only to count the star shapes. Etc. Adding another level of cognitive demand to your task would serve to strengthen the impact of your results long term.

Line 426: I am glad you bring up the Response pad. Please see my thoughts above regarding the keys characteristics and add some information about that into this paragraph.

Given your conclusions, is perhaps a solution to avoid spending too much time on simple RT tasks that might lead to fatigue down the road? To prevent fatigue, perhaps the majority of trials should be focused on dual tasks that disrupt cognitive processes the most? Especially for an fMRI response? Perhaps not. Could you please comment on that? Please comment on the implications of your study specifically for using in an fMRI study as well.

Reviewer #2: The tasks should be presented and described in Methods/Procedure subchapter instead in Introduction.

Why had healthy controls a higher MoCA threshold for the enrolment in the study? The explanation should be given for this different cognitive threshold.

This patient and on healthy subject, with MoCA score <21, should not be included, rather then be excluded.

Table 1. The t-test should be performed and p values given for possible difference in age, education or MoCA score, between patients and controls.

Why was it necessary to remind subject of dual task approaching?

At the beginning of last paragraph od chapter 2.3 the authors should give an explanation why exactly were participants given the questionnaire on whether they thought the trials in the SRTT followed a pattern?

The manuscript needs thorough English grammar & language editing.

6. PLOS authors have the option to publish the peer review history of their article (what does this mean?). If published, this will include your full peer review and any attached files.

Reviewer #1: **Yes: **Tiphanie E Raffegeau

Reviewer #2: No

---

## [Author Response · Author response to Decision Letter 0]

23 Mar 2021

Request: Please ensure that your manuscript meets PLOS ONE's style requirements, including those for file naming 

Response: We hope that all files are now named correctly

Request: Please amend your current ethics statement to include the full name of the ethics committee/institutional review board(s) that approved your specific study.

Response: the name provided "the regional Research Ethics Board of Stockholm" is the full name of the ethics committe/institutional and we have therefor not made changes in this regard.

Request: Please provide additional details regarding participant consent. 

Response: we now have added that consent from participants was written. 

Request: If there are no restrictions, please upload the minimal anonymized data set necessary to replicate your study findings as either Supporting Information files or to a stable, public repository and provide us with the relevant URLs, DOIs, or accession numbers. 

Response: the data is now openly availiable at https://osf.io/x9baq/ , DOI 10.17605/OSF.IO/X9BAQ.

Reviewer #1 

This study was a nice feasibility study that looked at implicit versus explicit motor learning in people with Parkinson’s Disease. To determine the influence of cognitive demands on motor learning, the authors included a dual task to a common key-press experiment. The study was intended to confirm the feasibility of using the task inside an fMRI machine. The authors have promoted transparency by providing their data and analysis, and used descriptive statistics to draw inferences because their study was admittedly underpowered.

I think this is a great feasibility study, and I think more studies like this should be published. I just have minor suggestions for clarity before publication. Also please note there are some grammatical errors throughout the manuscript that should be addressed by an editor and this reviewer will ignore.

Response We are happy to hear that the reviewer appreciates our efforts to evaluate task feasibility as well as our efforts for transparency. We also thank the reviewer for pointing out that the language needs improvement and we have now used professional writing assistance and hope that this has substantially improved the language of the manuscript. 

Comment #1

Line 37: RT hasn’t been previously defined

Response: Thanks for pointing this out, reaction time is now written out (line 38) and the abbreviation is introduced outside the abstract on line 219. 

Comment #2

Line 47: Are you referring to deficits in motor learning and dual-tasking specific to PD? Please clarify. There is a modern meta-analysis that you could cite in your intro:

Raffegeau, T.E., Krehbiel, L.M., Kang, N., Thijs, F.J., Altmann, L.J.P., Cauraugh, J.H., Hass, C.J., 2019. A meta-analysis: Parkinson’s disease and dual-task walking. Park. Relat. Disord. 62, 28–35. https://doi.org/10.1016/j.parkreldis.2018.12.012

Response: We added “in individuals with PD” to clarify that we are describing the deficits for this population specifically. We have also added a reference to the suggested meta- analysis, line 49. 

Comment #3

Line 50: The following sentence is vague, please clarify that you are referring to people with PD.

Response: Thanks for pointing this out, we now hope this to be clarified by the addition “…that individuals with PD experience”, line 50. 

Comment #4

Line 52: Please clarify what you mean when you say walking consists of a ‘series of complicated movements’. Adding a sentence or two would help clarify the rationale of the experiment. I believe the argument could be that movement becomes less automatic with PD and thus could cause interference in motor learning mediated by cognitive processes.

Response: We have now tried to better explain what a sequence of movement is, using the example of reaching for something, please see line 53-55. 

We also agree that it is good idea to clarify our interest in implicit motor learning. We have tried to do so by further stressing how individuals with PD have a plausible deficit in implicit learning and that this may explain parts of their motor deficits, see line 60-63. 

Comment #5

Line 79-83: Please clarify briefly that fMRI imposes limitations on cognitive test administration that this study is designed to overcome. A novice reader may be lost without a brief review of the fact that the key task has to be administered in an isolated way without the participant moving, etc.

Response: Thanks for pointing this out, we have now clarified this, see line 91-93. 

Comment #6

Line 101-102: Add medication state information to the Participants section 2.2. Please add if you controlled the study such that your testing time was during optimal medication states, i.e. 1-2 hours after taking their medication?

Response: We have moved this information from section 2.1 to 2.2 and agree it might be a more suitable paragraph for this information, see line 133-136. We chose to not use fixed time for testing in relation to the latest medication intake. The reason being that the participants differed in type of dopaminergic medication used and thereby in the medications’ duration of action i.e., the optimal medication states differed in time after intake and also in length, for the different participants. 

Comment #7

I cant quite tell by the figure but I believe by the description of the task is that the keys are button-press keys. This may seem quite minor, but this reviewer is curious is the keys were rigid enough such that the commonly reported hand tremors wouldn’t influence performance for a PD patient? How much force is required to register a result? Later, there is good detail about how they examined key presses and excluded what appeared to be random responses, my guess is that if the key were too easy to press, that a tremor could be registered as a response. A major strength of this design would be therefor the capability to prevent random key presses that are the result of bradykinesia or dyskinesia. For instance, a ‘touch pad’ key that doesn’t incorporate any haptic feedback may not be a good idea because it would pick up many more false positives. This may be worth mentioning in the Methods and Discussion.

Response: We agree that this is a very important question, and the reviewer is right in that button-press keys were used. Unfortunately, we have no exact figures as for the force needed to press the buttons, we however have tried out the buttons thoroughly ourselves, observed the participants during the training and listened to the participants’ opinions about how the response pads worked. Altogether, we believe that the buttons had a good balance between not registering very light presses, as caused by for example tremor, and not being too resistant, which could be equally problematic. We added a comment on this in the discussion, see line 490-495. 

The response pads also came with the benefits of being available both as a MR compatible set and as an equal set to be used outside the scanner i.e., as in the present study and for the training of tasks performed before entering the scanner in the RCT. The response pads were also not overly complicated but solely included the number of buttons needed for the tasks. 

Link to the response pads used: https://www.curdes.com/mainforp/responsedevices/buttonboxes/hhsc-2x2.html.

Comment #8

Later, given the interest in task fatigue, I found myself curious as to how long the protocol took and how long the break was, but I don’t see that information? Maybe I missed it, but highlighting that would help interpret your results.

Response: We thank the reviewer for pointing out this lack of information. The total length of the protocol ranged from 60 to 90 minutes depending on participant characteristics such as how easily they understood the tasks and the level of their motor impairment. We have added information on total protocol length as well as length of breaks in the paragraph “2.1. Procedure”, see line 109-112. 

Comment #9

Line 212: Starting this sentence with a letter and numbers is confusing, adding the word ‘R software’ to this sentence would help.

Response: This is a very valid point and we have now added “The software” before “R” to ease reading of the sentence, line 221-222. 

Comment #10

The section 2.5 is a nice detailing of the goals of the study. The Results are pretty hard to follow as is, but this numbering system helps. Can the authors please incorporate this numbering system to section 3.3? It might be nice to have a section 2.4.1 for SRRT descriptives and 2.4.2 for dual task descriptives, that would help the reader follow the corresponding Results in section 3.1 and 3.2.

Response: This is an excellent suggestion. We have now created 2.4.1. (line 223) and 2.4.2. (line 240) for the SRTT and the dual task, respectively. We have also incorporated section “3.3.1. Primary feasibility aspects” (line 360) and section “3.3.2. Expected patterns in the data” (line 408) and hope that this will increase the readability. 

Comment #11

If you have enough room it would be nice to have a table or figure that presents the survey results, rather than reading through a long section.

Response: This is also a very good suggestion and we have now replaced the paragraph with a table, see line 398, Table 2. 

Comment #12

A suggestion for all figures, which do a nice job of showing individual data points. I suggest you use the ‘jitter’ function in whatever program you are using so that your data points re randomly distributed rather than displayed in ‘rows’

Response: We have now changed the box plots of reaction time and accuracy of the button presses for both the SRTT and the dual task. These outcomes consist of continuous scales where all individuals differ to some extent, even if sometimes only minor. These plots now more properly reflect these individual differences and we have also included a jitter in the wide direction to avoid confusing overlaps of the data points. We thank the reviewer for pointing out the earlier unnecessary simplification. 

As for the two plots of the counting part in the dual task, the scale is however not continuous but discrete and only integer values within the ranges of the y-axes can be obtained. That some participants are on the same row therefor reflects that they scored exactly the same, and these points cannot be jittered in the height direction without a distorted presentation of the participants’ real values. We have however added a jitter in the wide direction, as for the other plots. We hope that these changes altogether make the plots more enjoyable. 

Comment #13

Line 394-396: I agree with this point and I think also that the rate of learning would be an interesting outcome for your paradigm. Perhaps moving away from magnitude based hypothesis testing to testing the slope of the change would be best, especially in complicated clinical samples.

Response: We are happy that the reviewer shares our interest in optimal analyses of task data. Learning rates and potential differences between groups or effects of interventions on learning rates, are definitely a very interesting aspect to look further into! 

Comment #14

Line 407-409: While I agree your counting task likely leads to sufficient cognitive demand for a feasibility study, you could more carefully load ‘cognitive processes’ rather than simply dividing attention by adding some sort of inhibition/switching component to the task, which would greater challenge cognitive control/executive functions. While the counting task does load simple cognitive processes, your results could be more robust to PD-related deficits if you incorporated broader networks of cognitive processing. For instance, changing the colour of the crosses periodically and instructing participants to only count Red crosses. Or changing the shape of the symbol and asking them only to count the star shapes. Etc. Adding another level of cognitive demand to your task would serve to strengthen the impact of your results long term.

Response: We sincerely value that the reviewer brought up this interesting question on task design. We do agree that it would be very interesting to use a cognitive task that taxes executive functions to an even greater extent. However, based on our experience of the participants’ reactions and understanding of the task when presented to them, we think it would be quite difficult to make the task further complex. We believe that a more complex task would increase the anxiety and even willingness to perform the task for at least some of the participants. A higher level of anxiety is something we firmly try to avoid, especially when the tasks are to be performed inside the MRI scanner. Increased anxiety could confound the performance outcomes, increase movement during data acquisition and thereby distort the data, and even cause doubt about entering the scanner and performing the task at all. We would also like to argue that the dual task as designed now, is a simplified but plausible proxy measure of many important day to day tasks e.g., walking while also concentrating on something else (“what to do for dinner”, “was I to turn left here to get to the shop?” etc.). See line 468-471 for an added discussion on this question. 

Comment #15

Line 426: I am glad you bring up the Response pad. Please see my thoughts above regarding the keys characteristics and add some information about that into this 

paragraph.

Response: This is indeed a valid point and we have added a brief discussion on the possibility of involuntarily button presses and our attempt to counteract these errors, see line 490 - 495. 

Comment #16

Given your conclusions, is perhaps a solution to avoid spending too much time on simple RT tasks that might lead to fatigue down the road? To prevent fatigue, perhaps the majority of trials should be focused on dual tasks that disrupt cognitive processes the most? Especially for an fMRI response? Perhaps not. Could you please comment on that? Please comment on the implications of your study specifically for using in an fMRI study as well.

Response: 

We thank the reviewer for bringing up a very interesting discussion point. Fatigue is indeed a potential problem that needs to be taken into great consideration when designing tasks and studies. To prevent fatigue, limitations are always needed to some extent and choosing the focus/foci of a task or a paradigm must of course be dependent on the primary research question. We have a great interest in dual task ability, but we are also very interested in motor learning and motor performance in a more general sense i.e., not solely motor tasks with an explicit addition of an extra task. Since there are empirical indications that implicit motor learning, as tested by our first task the SRTT, is impaired in individuals with PD and that such an impairment is also theoretically plausible given that implicit learning is dependent on one of the most affected brain structures in PD, the striatum, we believe that the SRTT is plausible task for us to prioritize. While keeping both tasks presented in the same order as here during the acquirement of brain activity data, we will however be careful to include a proper break between the two tasks to diminish task related fatigue. We will acquire structural brain imaging data in the same scanning session as the task performance and we will make sure to acquire this structural data in between the SRTT and the dual task. This will result in 7-10 minutes between the two tasks where the participants will be encouraged to relax both fingers and eyes. We believe that this is reasonable compromise between risk of task fatigue and efficient acquiring of research data. We also hope that this reasoning answers the reviewer’s quest for a discussion on specific implications for task use during the brain activity acquirement. See line 453-455 in the manuscript for a clarification on that we strongly recommend the use of proper break between the two tasks. 

Reviewer #2

Comment #1

The tasks should be presented and described in Methods/Procedure subchapter instead in Introduction.

Response: We thank the reviewer for bringing up the question of where the tasks should best be described. In line with the suggestion, we have moved the more specific and technical details of the tasks to the method section, see line 65-68 in the introduction and line 150-153 in the method section. We agree that it is more commonplace to fully save the description of any tasks used, for the methods section -in the common situation where the focus of the study is on the abilities measured by the tasks (here implicit learning and dual task performance) - rather than the tasks themselves. However, as the present study’s focus is entirely on evaluating the specific tasks, rather than the abilities they measure, we find it reasonable to introduce the tasks in the introduction. We sincerely hope and believe that the modification we have done is satisfactory for the reviewer as well as other readers. 

Comment #2: 

Why had healthy controls a higher MoCA threshold for the enrolment in the study? The explanation should be given for this different cognitive threshold.

Response: We are glad the reviewer brought this question up as we find that this discussion is of great importance for studies of both PD and healthy older individuals. 

A score of 26 on MoCA is a common cut-off for probable mild cognitive impairment. However, studies have convincingly showed that this cut-off is too strict for healthy, older individuals, resulting in erroneous conclusions about whether an individual suffers from mild cognitive impairment. A score of 23 has been reported as a more reasonable cut-off for mild cognitive impairment in healthy older individuals with a sensitivity of 0.96 (0.79–0.99) and a specificity of 0.95 (0.87-0.99) in a sample of individuals ≥70 (1). At the background of these excellent psychometric figures, we therefore used the cut-off of 23 on MoCA for our healthy participants. 

Cognitive deficits are frequent in individuals with PD and so to recruit samples to our studies that are representative of the PD population, we need to allow inclusion of participants with PD with a certain degree of cognitive impairment. However, by experience, we know that the participants in our studies cannot be too severely affected in cognition as they then will have a difficult time to participate and benefit from the training programs we investigate. Hence, we chose the cut-off criteria of MoCA to be 21 for our participants with PD. 

See line 130-133 for added information on this important question.

We hope that this explains our two chosen cut-off criteria in a satisfactory way. 

Comment #3

This patient and on healthy subject, with MoCA score <21, should not be included, rather then be excluded.

Response: We interpret this comment as that the reviewer suggests that participants scoring under our MoCA cut-off values should be described as “not included” instead of “excluded” and we have changed the phrasing in line with this suggestion. See line 136-138. We believe that providing information on excluded participants is an important contribution to the transparency of the research process. If our interpretation of the reviewer’s comment is not a correct understanding, please let us know. 

Comment #4

Table 1. The t-test should be performed and p values given for possible difference in age, education or MoCA score, between patients and controls.

Response: We have performed t-tests and reported p-values for age and education. See line 139-141. We have chosen to not t-test the difference in MoCA scores as this is a measure where we expect a difference between the groups due to common cognitive deficits in individuals with PD and also due to the difference in cut-off values for the MoCA. 

Comment #5

Why was it necessary to remind subject of dual task approaching?

Response: We appreciate that the reviewer brought up this important design question. In this type of task where a stimulus is repeated several times and the effect of the stimuli are averaged over the task, we want the effect of each singular stimulus to be as similar as possible. For the dual task specifically, we want the participants to experience a similar difficulty level or effect, every time the plus sign is flashed. To diminish a potential surprise effect of the first flashed plus sign, we choose to tell/show the participants when they should expect the dual task i.e., counting the plus signs. Yet another reason to avoid such a surprise effect is the possible accompanying movement. It is of great importance to minimize all movements when the task is to be performed inside an MRI scanner since any movement can have severe effects on the quality of the brain activity data. We have added this important information on line 183. 

Comment #6

At the beginning of last paragraph od chapter 2.3 the authors should give an explanation why exactly were participants given the questionnaire on whether they thought the trials in the SRTT followed a pattern?

Response: We thank the reviewer for pointing out this lack of information. We have now added such an explanation that we hope is satisfactory in its level of detail, see line 194 - 196. 

Comment #7

The manuscript needs thorough English grammar & language editing.

Response: We have now used professional writing assistance and hope that this has substantially improved the language of the manuscript. 

1. Luis CA, Keegan AP, Mullan M. Cross validation of the Montreal Cognitive Assessment in community dwelling older adults residing in the Southeastern US. Int J Geriatr Psychiatry. 2009;24(2):197-201.

---

## [Decision Letter · Decision Letter 1]

5 May 2021

Measuring implicit sequence learning and dual task ability in mild to moderate Parkinson's Disease: A feasibility study

PONE-D-20-28422R1

Dear Dr. Freidle,

We’re pleased to inform you that your manuscript has been judged scientifically suitable for publication and will be formally accepted for publication once it meets all outstanding technical requirements.

Kind regards,

Imre Cikajlo, Ph.D.

Academic Editor

PLOS ONE

Additional Editor Comments (optional):

Reviewers' comments:

Reviewer's Responses to Questions

**Comments to the Author**

1. If the authors have adequately addressed your comments raised in a previous round of review and you feel that this manuscript is now acceptable for publication, you may indicate that here to bypass the “Comments to the Author” section, enter your conflict of interest statement in the “Confidential to Editor” section, and submit your "Accept" recommendation.

Reviewer #1: All comments have been addressed

2. Is the manuscript technically sound, and do the data support the conclusions?

Reviewer #1: Yes

3. Has the statistical analysis been performed appropriately and rigorously? 

Reviewer #1: N/A

4. Have the authors made all data underlying the findings in their manuscript fully available?

Reviewer #1: Yes

5. Is the manuscript presented in an intelligible fashion and written in standard English?

Reviewer #1: Yes

6. Review Comments to the Author

Reviewer #1: The authors have done a nice job incorporating the reviewer's comments. After another round of grammar-checking and editing and this paper will make a nice addition to the literature.

7. PLOS authors have the option to publish the peer review history of their article (what does this mean?). If published, this will include your full peer review and any attached files.

Reviewer #1: **Yes: **Tiphanie E Raffegeau

---

## [Editor Report · Acceptance letter]

14 May 2021

PONE-D-20-28422R1 

Measuring implicit sequence learning and dual task ability in mild to moderate Parkinson´s disease: A feasibility study 

Dear Dr. Freidle:

I'm pleased to inform you that your manuscript has been deemed suitable for publication in PLOS ONE. Congratulations! Your manuscript is now with our production department. 

Kind regards, 

on behalf of

Professor Imre Cikajlo 

Academic Editor

PLOS ONE